# Untargeted Metabolomic Analysis of Lactation-Stage-Matched Human and Bovine Milk Samples at 2 Weeks Postnatal

**DOI:** 10.3390/nu15173768

**Published:** 2023-08-29

**Authors:** Dominick J. Lemas, Xinsong Du, Bethany Dado-Senn, Ke Xu, Amanda Dobrowolski, Marina Magalhães, Juan J. Aristizabal-Henao, Bridget E. Young, Magda Francois, Lindsay A. Thompson, Leslie A. Parker, Josef Neu, Jimena Laporta, Biswapriya B. Misra, Ismael Wane, Samih Samaan, Timothy J. Garrett

**Affiliations:** 1Department of Health Outcomes and Biomedical Informatics, College of Medicine, University of Florida, Gainesville, FL 32608, USA; xinsongdu@ufl.edu (X.D.); xu.ke@ufl.edu (K.X.); adobrowolski@ufl.edu (A.D.); magdafrancois@ufl.edu (M.F.); lathompson@peds.ufl.edu (L.A.T.); wanei@ufl.edu (I.W.); samih.samaan@ufl.edu (S.S.); 2Department of Obstetrics and Gynecology, College of Medicine, University of Florida, Gainesville, FL 32608, USA; jlaporta@wisc.edu; 3Center for Perinatal Outcomes Research, College of Medicine, University of Florida, Gainesville, FL 32608, USA; parkela@ufl.edu; 4Department of Animal and Dairy Sciences, University of Wisconsin-Madison, Madison, WI 53706, USA; bethany.senn@wisc.edu; 5Department of Behavioral Nursing Science, College of Nursing, University of Florida, Gainesville, FL 32603, USA; mmag@stanford.edu; 6Department of Physiological Science, Center for Environmental and Human Toxicology, College of Veterinary Science, University of Florida, Gainesville, FL 32608, USA; juan.henao@bpgbio.com; 7Division of Breastfeeding and Lactation Medicine, University of Rochester Medical Center, Rochester, NY 14642, USA; bridget_young@urmc.rochester.edu; 8Department of Pediatrics, College of Medicine, University of Florida, Gainesville, FL 32608, USA; neu@ufl.edu; 9Enveda Therapeutics, Inc., Boulder, CO 08301, USA; bbmisraccb@gmail.com; 10Department of Pathology, Immunology and Laboratory Medicine, College of Medicine, University of Florida, Gainesville, FL 32608, USA; tgarrett@ufl.edu

**Keywords:** human milk, breastfeeding, infant feeding, metabolomics

## Abstract

Epidemiological data demonstrate that bovine whole milk is often substituted for human milk during the first 12 months of life and may be associated with adverse infant outcomes. The objective of this study is to interrogate the human and bovine milk metabolome at 2 weeks of life to identify unique metabolites that may impact infant health outcomes. Human milk (*n* = 10) was collected at 2 weeks postpartum from normal-weight mothers (pre-pregnant BMI < 25 kg/m^2^) that vaginally delivered term infants and were exclusively breastfeeding their infant for at least 2 months. Similarly, bovine milk (*n* = 10) was collected 2 weeks postpartum from normal-weight primiparous Holstein dairy cows. Untargeted data were acquired on all milk samples using high-resolution liquid chromatography–high-resolution tandem mass spectrometry (HR LC-MS/MS). MS data pre-processing from feature calling to metabolite annotation was performed using MS-DIAL and MS-FLO. Our results revealed that more than 80% of the milk metabolome is shared between human and bovine milk samples during early lactation. Unbiased analysis of identified metabolites revealed that nearly 80% of milk metabolites may contribute to microbial metabolism and microbe–host interactions. Collectively, these results highlight untargeted metabolomics as a potential strategy to identify unique and shared metabolites in bovine and human milk that may relate to and impact infant health outcomes.

## 1. Background

Human milk (HM) is widely recognized as the optimal source of nutrition for infants due to its comprehensive nutritional composition in addition to hormones, enzymes, prebiotics, protective immunomodulators, and active microbiome [1]. Consumption of HM includes infant health benefits such as reduced risk of infections, respiratory diseases, diabetes, gastrointestinal diseases, diarrhea, and malnutrition, including both underweight and overweight [2,3]. Although little is known about breastfeeding and HM consumption in prehistoric humans, analysis of teeth collected from early humans has revealed that children routinely consumed HM up to six years after birth [4] and weaning likely occurred between 1–2 years of age [5]. In contrast, recent data from the United States Centers for Disease Control and Prevention (CDC) indicate that more than 8 in 10 (83.9%) U.S. infants will initiate breastfeeding shortly after birth; however, only about 1 in 4 (26.8%) of these children will continue to exclusively consume HM through the first 6 months of postnatal life [6]. Cow’s-milk-based infant formula is the most common substitute for mother’s milk in infancy when breastfeeding is impossible, undesired, or insufficient [7]. Accumulating data demonstrate that consumption of bovine whole milk during the first 12 months of life is associated with an increased risk of gastrointestinal bleeding [8,9,10,11], iron deficiency anemia [12,13,14], increased renal solute load [15], and allergies associated with feeding cow’s milk before 12 months [16]. Given that some infants will likely consume cow’s milk during the first year of life [7], there is an increasing need for information on cow milk composition to understand how the timing of exposure to cow milk’s bioactive compounds impact infant nutrition and health outcomes.

Historically, most milk composition studies have been performed in bovine samples using targeted chemical analysis aimed at characterizing specific classes of compounds (i.e., lipids, sugars, etc.) [8]. Cow’s milk is a rich source of proteins, lipids, vitamins, sugars, and minerals that are important for infant nutrition [9]. Accumulating data demonstrate that the nutritional and non-nutritional bioactive components of human and cow’s milk differ based on infant age [10], circadian rhythm [11,12], and stage of lactation [13]. Similarly, the composition of milk has been shown to vary based on nutritional intake [14], maternal diet [15], and geographic [16] and environmental influences [17]. Recent advances in mass spectrometry (MS) have increased the number of untargeted techniques available to characterize clinically relevant small molecules in biological tissues such as human milk [18]. The rapid growth in untargeted MS-based instrumentation has resulted in a proliferation of commercial and open-source software tools that can facilitate the discovery and identification of hundreds to thousands of metabolites in both clinical and research settings [19]. Despite these observations, variations in biological specimen collection, sample preparation, and analytical workflows have made comparisons between infant feeding studies difficult. As a result, population-based analysis of milk consumption during early life using untargeted MS-based methods has been limited to a small number of studies with small sample sizes.

The objective of this study is to interrogate the human and bovine milk metabolomes at 2 weeks postnatal to identify unique metabolites that may influence infant health outcomes. Given that the milk metabolome has been shown to vary within and between mammals according to clinical [16], geographical [16], and environmental factors [20], we matched human and bovine samples based on clinical and geographic factors to control for factors associated with variation in the milk metabolome. Moreover, we selected 2-week milk samples based on the premise that infants are more likely to receive human milk during the first months of life. Our analysis of lactationally matched milk samples has potential to inform the understanding of the unique and overlapping metabolomic profiles in human and bovine samples. Finally, we have linked milk metabolites with biological knowledge from the Human Metabolome Database [21,22,23,24,25] that includes chemical taxonomy, microbe–host interactions, and health outcomes.

## 2. Materials and Methods

Appendix A illustrates key resources used in this study, including the source and identifier information. Detailed information is illustrated below.

### 2.1. Milk Collection and Handling

Data for this study were collected as part of the Breastfeeding and Early Child Health (BEACH) study. Briefly, the BEACH study is a longitudinal cohort study focused on studying the impact of maternal obesity on the composition of human milk, the gut microbiome, and infant outcomes. Participants were enrolled in the BEACH study between 36–38 weeks of pregnancy as either normal weight (NW; pre-pregnant BMI < 25.0 kg/m^2^) or obese (Ob; pre-pregnant BMI > 30.0 kg/m^2^), and maternal–infant dyads were followed through the first 12 months of postnatal life. Pre-pregnant BMI was self-reported and validated by the research team through review of prenatal electronic health records. Human milk (*n* = 10) was collected at 2 weeks postpartum from normal-weight mothers (pre-pregnant BMI < 25 kg/m^2^) that vaginally delivered term infants and planned to exclusively breastfeed for at least 2 months. Participants were instructed to collect a fasted, morning, mid-feed HM sample as previously described [26]. In short, a feeding session was initiated by the infant and when the feed was halfway complete, the infant was removed from the breast, the breast cleaned with sterile water and gauze, and a sterile manual breast pump was used to express ~10 mL of HM. Milk was immediately placed on ice and transported to the laboratory where it was stored at −80 °C until analysis [27]. Similarly, approximately ~10 mL of bovine milk (*n* = 10) was collected 2 weeks postpartum from normal-weight primiparous Holstein dairy cows. Dairy cattle were housed in sand-bedded, shaded barns with access to fans and water soakers and fed a common transition cow total mixed ration. After milk sample collection, samples for metabolomics assays were thawed on ice, homogenized, aliquoted, frozen again with dry ice, and stored at −80 °C sample preparation for LC-MS/MS in analytical instruments. This study was approved by the Institutional Review Board of University of Florida and the Institutional Animal Care and Use Committee at University of Florida in accordance with the relevant guidelines and regulations. Written informed consent was obtained from each human participant.

### 2.2. Metabolite Extraction

All samples were extracted following previously published procedures that include pre-normalization according to protein content [28]. All samples were kept on ice throughout all steps and all reagents were LC-MS grade (Thermo Fisher Scientific, Waltham, MA, USA). Blank samples that contained deionized water were extracted following the same procedure with milk samples. In brief, to each 50 µL (normalized to 15,000 µg/mL protein) sample of milk, 10 µL internal standard solution (IS) was added. Spike-in IS mixture (Acros Organics, Fairlawn, NJ, USA) is described in Appendix A, which was dissolved in 0.1% formic acid in water. Samples were briefly vortexed after adding the IS mix, then 800 μL 8:1:1 acetonitrile:methanol:acetone was added for protein precipitation. Samples were vortexed and incubated on ice for 30 min. After incubation, samples were centrifuged at 20,000× *g* for 10 min at 4 °C to pellet the protein. Supernatants (750 μL) were transferred to new 1.6 mL vials, and dried under nitrogen at 30 °C. Once dried, samples were reconstituted in 100 μL 0.1% formic acid in water, vortexed, incubated on ice for 20 min, then centrifuged at 20,000× *g* for 10 min at 4 °C to pellet any remaining protein. Supernatants (80 μL) were removed and transferred to new 1.6 mL vials. From each vial, 10 μL was diluted with 90 μL 0.1% formic acid in water in glass LC–MS vials for data acquisition.

### 2.3. Analytical Instrumentation

Samples were analyzed by ultra-high-performance liquid chromatography–high-resolution mass spectrometry (UHPLC-HRMS), which is a popular technique for clinical metabolomics research such as drug surveillance [29], on a Q Exactive orbitrap mass spectrometer paired with a Dionex Ultimate 3000 UHPLC System (Thermo Fisher Scientific), employing a similar method to the work of Chamberlian et al. Briefly [28], metabolites were separated chromatographically on an ACE 18-pfp column (100 mm × 2.1 mm, 2.0 μm) (Advanced Chromatography Technologies, Ltd., Aberdeen, UK) using reversed-phase gradient elution (Solvent A: 0.1% formic acid in water, Solvent B: acetonitrile) at 0.35 mL/min with the following approach: 0–3 min: 100% A, 3–13 min: linear increase to 80%B, 13–16.5 min: 100% A, 16.5–20 min: 100%A at 0.6 mL/min (column flush and equilibration), 20–20.5: 100% A and 0.35 mL/min. The column temperature was held constant at 25 °C, and 2 μL for positive mode and 4 μL for negative mode were used as injection volumes. Positive and negative electrospray ionization at 35,000 mass resolution scanning from *m*/*z* 70–1000 was used as the method of full scan data acquisition. Source parameter settings were as follows: capillary voltage 3.5 kV; probe temperature 350 °C; capillary temperature 320 °C; sheath gas 40; auxiliary gas 10; sweep gas 1.0. Samples were run in one batch. Quality control (i.e., samples with internal standards and pooled samples) and blank samples were distributed evenly in the analytical batch.

### 2.4. Data Processing

Raw data files were converted to mzML format using ProteoWizard-msConvert [30]. The mzML files were then processed with MS-DIAL version 4.48 [31], which was used for signal processing. Parameters for data processing are included in Appendix A. Specifically, a minimum peak height of 10,000 along with a minimum peak width of 5 were implemented for peak detection; sigma window value of 0.5 and ESI spectra cutoff of 50,000 were used for deconvolution. With regard to metabolite annotation, an in-house built library from SECIM Core [32], together with publicly available libraries, including MassBank [33], Human Metabolome Database (HMDB) [25,26,27,28,29], MoNA (https://mona.fiehnlab.ucdavis.edu/, accessed on 26 April 2021), were integrated and used for MS1 and MS2 annotation, and the procedure of generating the merged libraries was based on previously published protocols [34]. Both positive and negative ion modes were implemented. Metabolite names were standardized with identifiers listed in HMDB. A detailed approach to aggregating a local version of public ESI spectral database is available [34]; 0.1 min of retention time (RT) tolerance, 0.01 Da of accurate mass tolerance, and 85% cut-off were used for MS1 annotation; 0.002 Da of accurate mass tolerance and 70% cut-off were used for MS2 annotation. We also assigned each identified metabolite a confidence level. Confidence level 1 is the highest level, which means we were very confident that our annotation was correct, and confidence level 4 is the lowest level. For peak alignment, the RT tolerance was 0.2 min and retention time factor was 0.5. In terms of noise removal, the comparison of 5 times of the averaged sample area with the averaged blank sample area was used for blank subtraction; peaks which existed in less than 20% samples or with a signal-to-noise ratio less than 5 were removed. The LOWESS algorithm was used for peak normalization. MS-FLO was used for post-processing including duplicate removal and isotope match and adduct match [35].

### 2.5. Statistical Analysis

Statistical analysis and visualization were carried out with R-4.0.2 [36]. Multiple statistical analyses were used to highlight the difference between two types of milk. Fold change was calculated by averaged intensity of bovine milk samples divided by that of human milk samples. Student’s *t*-test with an adjusted *p*-value (adjusted by Benjamini–Hochberg false discovery rate of 0.05) of 0.05, as well as absolute log2 fold change of 1, were used for significant metabolite identification [37]. We linked milk metabolites to disease outcomes in HMDB [21,22,23,24,25] and disease outcomes were condensed using the Human Disease Ontology [38]. We used Sankey plots to visualize pleiotropic metabolite–disease associations. Sankey diagrams are flow diagrams in which the arrow width is proportional to the quantity of metabolites linked to disease outcomes. Chemical enrichment statistical analysis was performed with ChemRICH [39],which is a chemical similarity-based statistical enrichment software with better subsequent enrichment statistics than pathway analysis and does not depend on biochemical knowledge annotations.

## 3. Results

### 3.1. Data Processing

Appendix A summarizes the results of computational data processing for human and bovine LC-MS/MS datasets. MS-DIAL analysis resulted in 4310 features for positive ion mode and 4054 features for negative ion mode analysis. After result optimization and deduplication with MS-FLO, the feature numbers were further reduced to 4070 (240 were removed) and 3910 (153 were removed) for positive and negative ion modes, respectively. Of the 240 positive ion features, 144 of them were matched automatically by MS-FLO while 96 were matched after manual review; of the 153 negative ion features, 103 were auto-matched while 50 were manually matched. Python scripts were used for noise removal and blank subtraction. After noise removal, there were 2280 features for positive mode and 2696 features for negative mode. After duplicate removal and biological likelihood filtering (i.e., HMDB identifier mapping), 155 unique identified positive features and 102 unique identified negative features were selected. Overall, 154 positive features were associated with a confidence level of 1 while 1 was associated with a confidence level of 3; 101 negative features were associated with a confidence level of 1 while 1 was associated with a confidence level of 3.

### 3.2. Differences between Human and Bovine Milk Metabolites

Figure 1A shows 46 metabolites that were significantly different between bovine (15 enriched metabolites) and human (31 enriched metabolites) samples. The most significant metabolites enriched in human milk included (2*R*)-6-methylpiperidine-2-carboxylic acid, cytidine, and 1,2-cyclohexanedione. The most significant bovine metabolites were indoxyl sulfate, *p*-cresol sulfate, and valine. Figure 1B shows a Venn diagram where 83% (*n* = 202) of milk metabolites with chemical taxonomy information belong to nine classes which were shared between human and bovine milk. The primary classes shared among milk samples included amino acids, lipids, and xenobiotics. We found that human milk samples were enriched with 30 metabolites that had known classes, and the 30 metabolites belonged to eight HMDB classes that included organic acids and derivatives, lipids and lipid-like molecules, and organic oxygen compounds. Bovine milk samples were enriched with 15 metabolites that had known classes, and the 15 metabolites belonged to five classes, including organic oxygen compounds, nucleosides, and benzenoids. Figure 1C shows that an unsupervised principal component analysis (PCA) revealed that two groups of samples were different. Figure 1D shows the distribution of metabolites with statistical significance across the mass range, generally with a *m*/*z* value of less than 400. We found that 60% (155) of metabolites were detected in positive ionization mode and 40% (102) were detected in negative ionization mode, indicating the importance and complementarity of both modes of data acquisition. Figure 1E shows the distribution of metabolites with statistical significance across retention time and revealed that most metabolites had a very short retention time, which are expected to be sugars.

Figure 2A presents a pie chart with the overall composition of the milk samples collected from bovines and humans. A pie chart is a circular graphical representation used to display data as a simple way to visualize the composition of different parts or categories within a whole. An overall breakdown of the composition of the pie chart is shown in Appendix A. Briefly, we found that the metabolomic composition of human and bovine milk is dominated by amino acids (23.7%), xenobiotics (15.18%), nucleotides (11.67%), carbohydrates (9.73%), and lipids (8.56%). We also detected low abundance (<5%) of metabolites related to vitamins and cofactors (3.5%), energy metabolism (3.89%), and peptides (1.96%). Notably, we found that approximately 20% of metabolites detected in milk samples had a relative abundance <1% at the sub-class level (Appendix A). Figure 2B presents a Sankey plot that shows the classification information of all identified metabolites and their associations with linked diseases in the HMDB database. A Sankey plot is a specific type of data visualization that illustrates the flow of resources, and it is particularly effective for visualizing the distribution and allocation of data. We found that amino acids had the largest proportion of metabolite–disease links that were associated with the gastrointestinal system, inherited metabolic disorders, and the nervous system.

Figure 3 shows that unsupervised hierarchal clustering of milk samples clearly differentiated human and bovine samples. Hierarchical clustering plots provide insights into the structure of the data and how different data points or clusters relate to each other. Overall, we found 46 metabolites across eight classes that were significantly different between human and bovine samples. Specifically, we found that the most significant metabolites belong to amino acids and xenobiotics. According to the bar plot on the right side of Figure 2, Rhamnose had the highest significant level (i.e., lowest FDR), and tartaric acid and succinylcarnitine had the highest fold change. Figure 4 includes Sankey plots for metabolites enriched in bovine milk (A) and enriched human milk (B) and their individual associations with diseases. We found that most bovine-milk-enriched metabolites belonged to the nucleotide class and were related to gastrointestinal system disease and cancer. Appendix A presents an enrichment analysis that revealed four significant clusters (raw *p* < 0.05) in total, among which piperidines were the most significant, and enriched in bovine milk.

## 4. Discussion

### 4.1. Overview

Human breast milk is recognized as the ideal nutrition source for infant development [18,40]; however, many infants consume some cow’s milk during their first year of life. Saldan et al. demonstrated that cow’s milk was consumed by 16% of infants in the first month of life, 50% of infants through 6 months, and 90% of infants by 12 months of life in a Brazilian cohort [41]. These results are generally consistent with data from Mexico that 7.5% of infants 4–6 months old consumed some whole cow’s milk, and by 12 months of age, more than 30% of infants were consuming some cow’s milk [42]. In the United States, cow-milk-based commercial infant formula, which is a modification of cow milk, is the predominant human milk substitute that is fed to infants who are unable to receive human milk [7]. Given that some infants will likely consume cow’s milk during the first year of life [41], there is a need for information on milk composition to understand how the timing of exposure to bovine milk metabolites impact infant nutrition and health outcomes. The objective of this study is to interrogate the human and bovine milk metabolome at 2 weeks postnatal to identify unique metabolites that may impact infant health outcomes. Our LC-MS workflow was able to detect 256 milk metabolites that included 83 (32%) milk metabolites that have not previously been reported using untargeted approaches. Analysis of microbial-derived metabolites revealed that nearly 80% of milk metabolites identified in our study may contribute to microbial metabolism and microbe–host interactions. Finally, our analysis of metabolite–disease associations revealed that milk metabolites have a pleiotropic impact on health outcomes. Collectively, our results raise the possibility that the adverse health outcomes that result from consumption of cow’s milk during early life may be mediated by microbe–host interactions with bovine metabolites that are not present in human milk.

### 4.2. Previous Work

Analysis of milk composition using untargeted MS-based methods has largely focused on metabolomics comparisons within bovine and human samples [8]. The most comprehensive assessment bovine milk was published in 2019 by Foroutan et al., where investigators combined nuclear magnetic resonance (NMR) spectroscopy, liquid chromatography–mass spectrometry (LC–MS), and inductively coupled plasma–mass spectrometry (ICP–MS) to quantify 296 milk metabolites from commercial milk samples and identified 676 metabolites through a “bibliometric analysis” of the published literature [8]. Qian et al. reported that non-esterified fatty acids were much more abundant in human milk than those in formula and bovine milk, and that levels of tricarboxylic acid (TCA) intermediates were much higher in formula and bovine milk relative to human milk [43]. More recently, Baumgartel et al. published a scoping review on the human milk metabolome that included 22 studies that used untargeted metabolomics techniques [44] and reported aggregate statistics for the number of metabolites identified as well as the geographic location, research design, analyses, and platform used [44]. A strength of our study is the direct comparison of lactationally matched and untreated human and bovine samples. Consistent with other studies [45], our results revealed that more than 80% of the milk metabolome is shared between human and bovine samples during early lactation. Though there is a core milk metabolome, exploring differences in the metabolites of human milk and comparing the metabolites to whole bovine milk may identify potential pathways to deepen our understanding of the protective mechanisms of human milk for human infants. Moreover, 67.7% of metabolites detected in our study have been previously identified in untargeted metabolomics studies, including 46 related to amino acid metabolism [8,45,46,47,48,49,50,51,52,53,54,55,56,57,58,59], 23 carbohydrates [46,47,50,51,52,53,55,59,60,61,62], 11 co-factors and vitamins [46,47,52,55,61,63], 26 nucleotides [45,46,47,48,49,50,54,55,57,58,60,61,62], and 21 xenobiotics [45,46,47,51,55,60,61,64,65]. An important observation from our study is the identification of 18 milk metabolites that were significantly different between human and bovine samples and have not been previously reported in the untargeted metabolomics literature. Given the connection between the bovine-specific metabolites and 10 disease pathways, our results highlight future areas of research to inform the health risks of cow’s milk consumption during infancy and potential mechanisms whereby cow’s-milk-based infant formula can be improved.

### 4.3. Microbe–Host Interactions

Microbe–host interactions describe molecular and physical interactions between the microbiota and the host [66]. Previous work has shown that human milk metabolites are independently associated with beneficial microbial metabolic pathways predicted to increase intestinal barrier function and reduce intestinal inflammation [26]. More recently, an increasing number of clinical trials are focused on harnessing microbial metabolites as therapies for various microbiome-related diseases [67]. Despite these observations, there is limited research focused on characterizing microbial-derived metabolites in milk and their potential impact on infant health outcomes [68]. Previous work in human milk has largely focused on human milk oligosaccharides (HMOs) as prebiotics in HM [69]. Our study did not detect HMOs, likely due to the fact that most HMOs were outside the mass range of the instrument; however, our analysis did reveal that nearly 80% of milk metabolites identified in our study may contribute to microbial metabolism and microbe–host interactions. Notably, we found several metabolites enriched in human samples that can participate directly in microbe–host interactions, including bile acids (allocholic acid), biopterins, betaine, and xenobiotics (tartaric acid). Human milk bile acids are potent antimicrobials and play an important role in the innate immune defense within the intestine [70]. Similarly, biopterins concentrations in human milk have been shown to increase rapidly during the first 2 weeks of life and animal models have revealed that biopterin synthesis regulates the pathogenesis of neonatal bacterial meningitis [71]. Human milk betaine has been associated with infant growth in two independent and geographically different cohorts [72]. Moreover, animal models have demonstrated that higher betaine intake during lactation increased milk betaine content in dams, modified the microbiome, increased intestinal goblet cell content, and led to lower adiposity and improved glucose homeostasis in adult offspring [72]. Previous data demonstrate that more than 80% of tartaric acid is consumed by the intestinal microbiota [73]. Animal models suggest that tartaric acid in milk impacts small intestinal and circulatory metabolome patterns observed in human milk (HM) feeding compared with cow’s milk formula (CMF) [74]. Collectively, these data demonstrate a link between human and bovine breast milk metabolites and microbe–host interactions that have potential to impact infant health outcomes.

### 4.4. Strengths, Limitations, and Future Works

We have included lactationally age-matched milk samples to understand the unique and overlapping metabolomic profiles in human and bovine samples. A strength of our study is the rigorous computational checklists [75] and data processing protocols [19] that were used to improve the transparency and reproducibility of our milk metabolomics workflow. Specifically, we have included all curated parameters for MS-DIAL and MS-FLO data processing in Appendix A and all identified peaks were also manually checked and curated to ensure their accuracy. Additionally, we have published the number of peaks produced by each step of the data processing protocol (Appendix A) and used an internal library which ensures high confidence level identification, as well as a good coverage of metabolites. Nevertheless, there are some limitations. The selection of peaks produced by MS-FLO were reviewed and curated by one researcher, which may introduce subjective bias. Although we merged several large and popular MS2 metabolite databases, the list of databases used for the merging might still not cover everything. Additionally, the sample size in this study is small, including only 20 samples, which is partly due to the expensive cost of metabolomics data acquisition. Moreover, it is unlikely that infants in the United States would have dietary exposure to 2-week postnatal bovine milk, but rather infant formulas based on cow milk from different lactation stages. Collectively, future investigation of the milk metabolome should consider implementing complementary analytical strategies (lipidomics, proteomics, metabolomics) in combination with an experimental design with an expanded evaluation of bovine-based infant formulas and bovine milk samples across a range of geographic and clinical factors. We also detected a larger number of xenobiotics in both human and bovine samples. However, we did not quantify the physiological concentrations of these compounds that would provide information on the potential bioactivity of these compounds. Nevertheless, these results highlight the potential for high-resolution mass spectrometry to detect compounds that reflect maternal dietary and environmental exposures. Given that both human and bovine samples were collected in Gainesville, Florida, we hypothesize that a shared environment is the primary explanation for xenobiotics or non-endogenous compounds that were shared across milk samples. Collectively, future investigation of the milk metabolome should consider implementing complementary analytical strategies (lipidomics, proteomics, metabolomics) in combination with an experimental design with an expanded evaluation of bovine-based infant formulas and bovine milk samples across a range of geographic and clinical factors.

## 5. Conclusions

While human milk is the optimal source of nutrition during early infancy, the majority of infants do not meet breastfeeding recommendations. In this study, we used an untargeted metabolomics data analysis technique to study the difference between human milk and bovine milk. Our analysis revealed connections between bovine-specific metabolites and health outcomes. Our results revealed that more than 80% of the milk metabolome is shared between human and bovine milk samples during early lactation. Unbiased analysis of identified metabolites revealed that nearly 80% of milk metabolites may contribute to microbial metabolism and microbe–host interactions. Collectively, these results highlight untargeted metabolomics as a potential strategy to identify unique and shared metabolites in bovine and human milk that may relate to and impact infant health outcomes.

## Figures and Tables

**Figure 1 nutrients-15-03768-f001:**
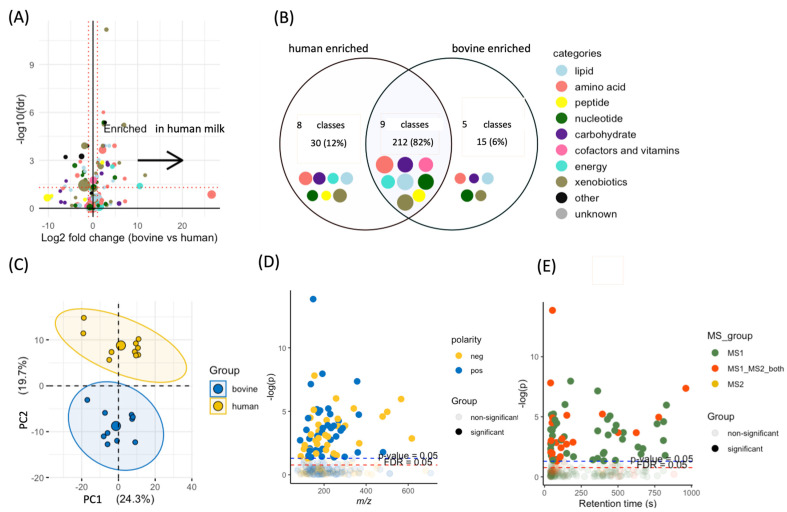
Statistical analysis of identified metabolites. (**A**) The volcano plot of identified metabolites. The sizes of dots stand for the averaged intensity of a metabolite, and the color is the metabolite’s class in HMDB. (**B**) The Venn diagram of identified metabolites that have a known class. (**C**) The PCA plot of bovine and human milk samples. (**D**) The Manhattan plot where the *x*-axis represents the *m*/*z* values of identified metabolites and *y*-axis represents −log(*p*) of metabolites. (**E**) The Manhattan plot of identified metabolites where the *x*-axis represents retention time of metabolites, and the *y*-axis represents −log(*p*) of metabolites.

**Figure 2 nutrients-15-03768-f002:**
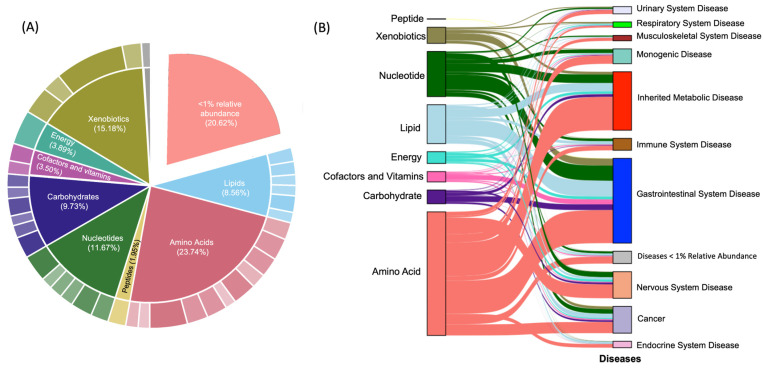
Pie chart and Sankey plot for significant metabolites of overall composition. (**A**) Pie chart. Indicates the 10 classes of metabolites discovered in both human and bovine milk samples. Amino acids were found to be the most enriched (23.74%). Color intensities of subsections within the pie chart indicate overall abundance of each subclass of metabolite found. (**B**) Sankey plot. Indicates associations of the 8 found metabolite classes (**left panel**) with 11 linked disease types (**right panel**). Each line between the two panels shows significant correlation with each other. The class of amino acid is seen to have the most abundant associations with diseases, specifically inherited metabolic disease and gastrointestinal system disease.

**Figure 3 nutrients-15-03768-f003:**
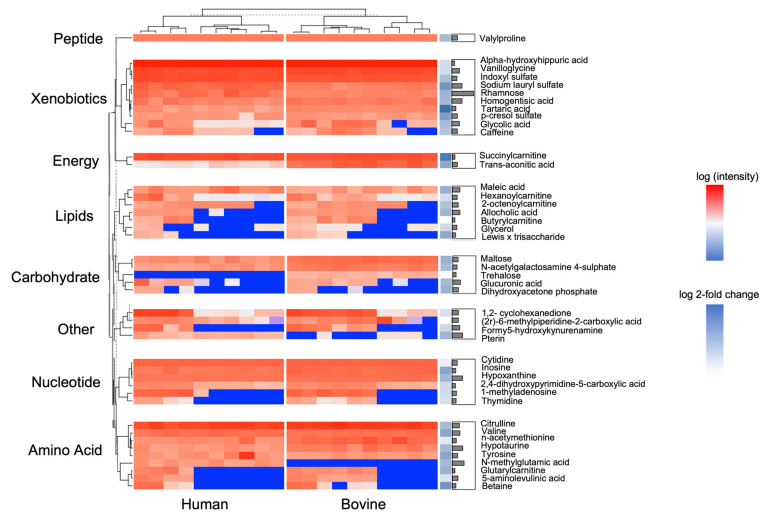
Heatmap for identified significant metabolites. Rows are separated by classes in HMDB and columns are separated by sample groups. The main heatmap scale ranges from −10 to 20 on a log2 scale. Red means high intensity while blue indicates low intensity. Log2 fold change is represented by the single-column heat bar on the right side of the map, and the bar plot on the rightmost section represents significant level (SL = −log10(FDR)). Annotations on the left side are classes in HMDB while those on the right side are metabolite names.

**Figure 4 nutrients-15-03768-f004:**
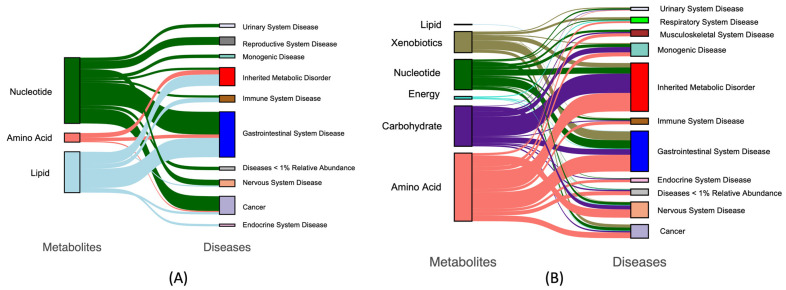
Sankey plots for significant metabolites. (**A**) Unique bovine metabolite associations. Indicates 3 significant metabolite classes (**left panel**) found within only the bovine samples that link to 10 disease types (**right panel**). The class nucleotide was found to be the most abundant with associations with gastrointestinal system disease and cancer. (**B**) Unique human metabolite associations. Shows 6 significant metabolite classes found only in the human samples (**left panel**) and their links to 11 disease types (**right panel**). The class of amino acids was the most abundant, with associations with inherited metabolic disorders and gastrointestinal disease.

## Data Availability

Data is available upon request.

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
