# Peer review of "Untargeted Metabolomic Analysis of Lactation-Stage-Matched Human and Bovine Milk Samples at 2 Weeks Postnatal"

_nutrients, 2023, doi:10.3390/nu15173768_

Round 1

Reviewer 1 Report

1.    Title, “Gestationally Matched 2 Human and Bovine Milk samples” Do the author mean lactation stage matched milk samples?

2.    What are the adverse outcomes when bovine whole milk is substituted for human milk during the first 12-months of life?  

3.    How were the milk samples collected? What was the detailed procedure and what was the volume of the collected milk from each subject or cow?

4.    The authors used pre-pregnant BMI as an index to recruit subjects for the study. Did the authors check the pregnant BMI for each subject?

5.     Why 2-weeks postnatal milk samples were chosen to conduct metabolomic analysis for human and bovine milk? It is unlikely that postnatal 2-week infants would have dietary cow milk. When bovine milk was used for infant formula manufacture, is cow milk from different stages or only mature bovine milk used to make infant formulas?

6.    Line 47, six years?

7.    Has any metabolomic analysis been done to compare cow milk and cow milk base infant formulas?

8.    Line 113, what are blank samples? Lines 110-119, were whole milk samples used for metabolite extraction?

9.    For data processing, is a bovine metabolome database used?

Reviewer 2 Report

Technically difficult study with resultant data of interest to a wide audience. I'd like a bit more detail on the collection/storage of human milk. Did you differentiate fore- from hind-milk? AM from PM collections? Was the milk homogenized to assure representative sampling? How big a sample was collected from each of the 10 human subjects?

Also, interested in why you chose 2-wk post-partum primip cows? What was logic/thinking behind this? Is this representative of the kind of cow milk that goes into formula production? Bottles of milk at Piggly Wiggly?

Xenobiotics are a major component in both human & bovine samples. Is that surprising? Can you comment on them? Discuss why they are there, where they came from, etc.?

Finally, you uses several different methods to plot the results. Many are not commonly used by clinicians. It might be helpful to give more information on how to interpret these plots.

Please see attached file for detailed comments.

English is good. I think I found a typo or 2.
